# Disulfide Bond Engineering of an Endoglucanase from *Penicillium verruculosum* to Improve Its Thermostability

**DOI:** 10.3390/ijms20071602

**Published:** 2019-03-30

**Authors:** Anna Bashirova, Subrata Pramanik, Pavel Volkov, Aleksandra Rozhkova, Vitaly Nemashkalov, Ivan Zorov, Alexander Gusakov, Arkady Sinitsyn, Ulrich Schwaneberg, Mehdi D. Davari

**Affiliations:** 1Federal Research Centre «Fundamentals of Biotechnology» of the Russian Academy of Sciences, Moscow 119071, Russia; chekushina.ann@gmail.com (A.B.); palman2008@yandex.ru (P.V.); inzorov@mail.ru (I.Z.); apsinitsyn@gmail.com (A.S.); 2Institute of Biotechnology, RWTH Aachen University, Aachen 52074, Worringerweg 3, Germany; s.pramanik@biotec.rwth-aachen.de (S.P.); u.schwaneberg@biotec.rwth-aachen.de (U.S.); m.davari@biotec.rwth-aachen.de (M.D.D.); 3G.K.Skryabin Institute of Biochemistry and Physiology of Microorganisms, Russian Academy of Sciences, Pushchino 142292, Moscow region, Russia; vnemashkalov@gmail.com; 4Department of Chemistry, M.V.Lomonosov Moscow State University, Moscow 119991, Russia; avgusakov@enzyme.chem.msu.ru; 5DWI-Leibniz Institut für Interaktive Materialien, Forckenbeckstrasse 50, Aachen 52056, Germany

**Keywords:** cellulase, endoglucanase, rational design, protein engineering, disulfide bonds, thermostability, cellulose biodegradation

## Abstract

Endoglucanases (EGLs) are important components of multienzyme cocktails used in the production of a wide variety of fine and bulk chemicals from lignocellulosic feedstocks. However, a low thermostability and the loss of catalytic performance of EGLs at industrially required temperatures limit their commercial applications. A structure-based disulfide bond (DSB) engineering was carried out in order to improve the thermostability of EGLII from *Penicillium verruculosum*. Based on in silico prediction, two improved enzyme variants, S127C-A165C (DSB2) and Y171C-L201C (DSB3), were obtained. Both engineered enzymes displayed a 15–21% increase in specific activity against carboxymethylcellulose and β-glucan compared to the wild-type EGLII (EGLII-wt). After incubation at 70 °C for 2 h, they retained 52–58% of their activity, while EGLII-wt retained only 38% of its activity. At 80 °C, the enzyme-engineered forms retained 15–22% of their activity after 2 h, whereas EGLII-wt was completely inactivated after the same incubation time. Molecular dynamics simulations revealed that the introduced DSB rigidified a global structure of DSB2 and DSB3 variants, thus enhancing their thermostability. In conclusion, this work provides an insight into DSB protein engineering as a potential rational design strategy that might be applicable for improving the stability of other enzymes for industrial applications.

## 1. Introduction

Cellulases are highly attractive enzymes for various industrial applications, such as lignocellulosic biomass conversion, cotton and paper manufacturing, juice extraction, and use as detergent enzymes and animal feed additives [1,2,3]. Cellulases are classified into three main groups according to their activities, including endo-1-4-β-glucanases (or simply endoglucanases), exo-cellobiohydrolases, and β-glucosidases. Endoglucanases cleave cellulose molecules internally, thus exposing new chain ends to the processive action of exo-cellobiohydrolases, which release cellobiose molecules into the solution. β-Glucosidases finally hydrolyze cellobiose to glucose. Endoglucanases have drawn much attention from researchers as an essential component of cellulase multienzyme cocktails for biomass degradation or as individual biocatalysts in some abovementioned applications [4,5,6,7].

The practical applications of endoglucanases are often limited because of the loss of enzymatic activity at industrially required temperatures. The increase in temperature by a few degrees above the enzyme temperature optimum leads to structural changes and the unfolding of the protein, thus affecting its function [8,9]. Therefore, the enzyme thermal stabilization is an essential requirement for its efficient use in biotechnology [10]. Protein engineering holds a potential to develop thermostable enzymes as biocatalysts. 

Recent advances in enzyme engineering through directed evolution [11] and “KnowVolution” [12] have successfully been applied to improve the thermostability of several enzymes, including cellulases [13,14], phytase [15], xylanases [16], and lipase [17]. However, using only the experimental techniques for the engineering of the enzyme stability is rather costly and time-consuming. Computer-assisted rational design is an attractive alternative to accelerate the enzyme engineering process. In the context of computational design to improve the thermostability of a biocatalyst, several approaches are widely used, such as the consensus sequence (comparing amino acid sequences with a higher thermostability), B-factor/RMSF analysis, molecular dynamics (MD) simulations, constraint network analysis (CNA), disulfide bonds (DSBs) design, and stabilizing salt-bridge design based on the sequence and structure of enzymes [18,19,20,21,22,23].

DSBs play an important role in protein folding and stability [24], enhance the stability of proteins that function in a fluctuating cellular environment, and contribute to the activity of many proteins by stabilizing them in their active conformations [25,26]. In recent years, a few studies have applied the DSB design approach to improve the thermostability of various enzymes, such as alkaline α-amylase [27], cellulases [28], and amadoriase [29]. Recently, the DSB engineering (N31C-T187C/P102C-N125C) of a mesophilic β-glucanase from *Bacillus terquilensis* showed an enhancement of the thermostability by 48.3% in increasing the half-life at 60 °C and a 4.1 °C rise in melting temperature compared to wild type [22]. However, DSB engineering to improve the thermostability remains to be unexplored for endoglucanase II (EGLII) from *P. verruculosum.*

In this work, we used the DSB engineering approach to improve the thermostability of EGLII from *P. verruculosum* belonging to the glycoside hydrolase family 5 [30,31]. EGLII is one of the major enzymes of the *P. verruculosum* extracellular multienzyme system, and it is an important component of the commercial enzyme preparation Agrocell Plus produced by the Agroferment company (Russia) for using as a feed additive. Although EGLII is one of the most thermostable enzymes of the *P. verruculosum* cellulase complex [30], its thermostability still needs to be improved since the industrial production of the animal feeds involves the pelletizing stage proceeding at approx. 80 °C. The beneficial variants of EGLII were chosen by computer simulation studies, allowing the characterization of the stability of the enzyme-engineered forms. Identified residue pairs were subjected to site-directed mutagenesis to generate DSBs. This report demonstrates that DSB engineering is an efficient rational designing approach for stabilizing enzymes in biomass bioprocessing applications. 

## 2. Results

DSB engineering was performed as an emerging approach to increase the thermostability of EGLII (Cel5A) from *P. verruculosum*. The results obtained are divided into three parts. In the first part, we describe the design and selection of DSBs obtained from Schrödinger’s BioLuminate software [32]. Secondly, we provide experimental data demonstrating that the engineered EGLII variants exhibited a higher thermostability in comparison with the wild-type enzyme (EGLII-wt). Finally, we explain the effect of DSBs on the overall enzyme structural stability, as well as the stability of DSBs, using MD simulations.

### 2.1. Computational Design and Screening of Disulfide Bonds

A native EGLII contains one DSB between C221 and C258. To increase the enzyme thermal stability, an introduction of extra DSBs into a protein structure was proposed. Based on the results of calculations with the BioLuminate software [32], three pairs of amino acid substitutions that could result in the formation of new DSBs were chosen (Figure 1). The substitutions of amino acid residues S12 and A270 (S12C-A270C, DSB1) take into account natively unfolded regions from the N- and C-termini of the protein. A disordered region in the middle of EGLII (amino acid residues 212–219) could not be used for mutagenesis because of its proximity to the enzyme active site. Therefore, a location close to this region but not affecting the active site and presumably capable of introducing the additional DSB (substitutions Y171C-L201C, DSB3) was chosen. The variant S127C-A165C (DSB2) was consistent with the parameters, except that the residues chosen for mutations were not part of the natively disordered regions.

### 2.2. Production, Purification, and Biochemical Characterization of the EGLII Variants

Genes encoding EGLII and its mutant forms carrying the abovementioned substitutions were cloned into the *Penicillium canescens* PCA-10 (*niaD*-) auxotrophic strain and was expressed under the control of a *xyl1* gene promoter. Recombinant EGLII-wt and its DSB2 and DSB3 variants were successfully expressed and then purified for a characterization using the sequential anion exchange and hydrophobic-interaction chromatography. However, the expression of the DSB1 variant was not successful, although a sequencing of the corresponding gene confirmed the presence of the target mutations. Figure 2 shows the SDS-PAGE of proteins from the culture fluids expressed by recombinant strains (**a**) and the purified EGLII forms (**b**).

The enzymatic activities toward carboxymethylcellulose (CMC) and barley β-glucan were determined for both the mutant forms of EGLII (DSB2 and DSB3) and the wild-type enzyme (Table 1). The engineered variants of EGLII exhibited higher specific activities against both substrates (by 15–21%) compared to the EGLII-wt. 

### 2.3. Thermostability of Recombinant EGLII Forms

To determine the thermostability of the EGLII recombinant forms, the enzyme solutions were incubated in a thermostat at 70 and 80 °C, and the residual activity against CMC and β-glucan was assayed after different times of incubation. The choice of conditions for the study of thermal stability was due to the fact that the EGLII is an important component of an industrial enzyme preparation for use as a feed additive (see Introduction), which the production of involves the pelletizing stage proceeding at approx. 80 °C. The inactivation curves for recombinant EGLII-wt and its DSB2 and DSB3 variants are shown in Figure 3 and Figure 4.

After incubation at 70 °C for 2 h, the engineered variants retained 52–58% of their activity toward both substrates, while the EGLII-wt retained only 38% of its activity. At 80 °C, the DSB2 and DSB3 forms retained 15–22% of their activity after 2 h, whereas EGLII-wt was completely inactivated after the same incubation time.

### 2.4. Analysis of Structural Stability

To understand the molecular mechanism of the increased thermostability caused by the introduction of DSBs, changes in the protein flexibility were analyzed by MD simulations. The MD simulations of EGLII-wt, DSB2, and DSB3 at 26.85, 70, and 80 °C showed that the DSBs in the engineered variants improved the thermostability by increasing the overall compactness of structures. This observation was evaluated by analyzing two key parameters, including root mean square deviation (RMSD) and (radius of gyration) R_g_. A higher RMSD typically indicates a more flexible and potentially less thermostability of the overall structure. The RMSD fluctuation remained quite similar in all EGLII forms at 26.85 and 70 °C; however, the engineered variants showed a higher stability at 80 °C in comparison with EGLII-wt (Figure 5). A similar phenomenon was observed in analyzing the R_g_ of the enzyme forms. At 80 °C, the structure compactness was retained in the DSB2 and DSB3 variants unlike in the EGLII-wt, implying that the DSBs increased the compactness of the enzyme (Appendix A).

As the designed DSBs improved the thermostability of the engineered variants, we evaluated the stability of the -S-S- distances (<2.15 Å) [33] through MD simulations. The distribution of C127_SG_-C165_SG_ and the C171_SG_-C201_SG_ distances were investigated to assess their stability during 100-ns simulations as retain in naturally occurring disulfides of enzymes (Appendix A). In the case of both DSB2 and DSB3 variants, the analyzed distances were retained within 2.15 Å in 200 data points of 100-ns simulations (Appendix A). Thus, the simulation results showed that the designed DSBs remained stable throughout the 100 ns. 

## 3. Discussion

Due to the covalent nature, the DSBs have strong effects on the folding and stability of a protein [34]. Nature has created the DSBs as a strategy to stabilize a protein in both intracellular and extracellular environments [25,35]. The effects of stabilization are expected to exert from the cross-linking of cysteine residues separated by a large number of intervening amino acids along the polypeptide chain [35]. Naturally, the DSB formation between cysteine residues occurs during the folding of proteins in their secretory pathway catalyzed by members of the protein disulfide isomerase family [36]. The introduction of extra DSBs through protein engineering without affecting the catalytic efficacy of enzymes was considered to improve the thermostability of mesophilic enzymes [22,37]. Therefore, in this study, we used rational DSB engineering to improve the thermostability of the EGLII from *P. verruculosum*.

Protein engineering based on directed evolution [11] and/or “KnowVolution” [12] has successfully been applied to improve the thermostability of different enzymes, although these techniques remain costly and time-consuming. Because of the rapidly increasing demand of highly thermostable enzymes, computer-assisted rational design and its experimental validation are desired. In the area of structure-based engineering, the DSB engineering offers several important advantages over the creation of extra hydrogen bonds, salt bridges, and hydrophobic cores because the DSBs are structurally well-defined and they are easily characterized [19,38]. The selection of the appropriate residue pairs for DSB introduction remains challenging as the new DSBs sometime impair enzyme activity while not contributing much to protein stability [22,32,39].

In this work. we applied a rational design, screening, and ranking of DSBs considering the energy ΔEi, ΔE, and the Quality parameter [32]. In the DSB1, S12 is located in the N-terminal and A270 is located in the α-helix region; however, this variant did not show an expression in the *P. canescens* host. The residue pair in the DSB2 is located in the α-helix-loop transition, whereas that in the DSB3 is located in the β-sheet-loop transition regions. The results from the MD simulation studies showed that the introduction of the DSB2 and DSB3 rigidified the global structures of the variants at higher temperatures. The reduction of the overall flexibility enhanced the protein thermostability. Moreover, both DSBs remained structurally stable and contributed to an improved thermostability though rigidifying the overall structures. 

The half-life times of EGLII at 70 and 80 °C increased by 1.5–2 times as a result of introducing the DSB2 and DSB3. A similar 1.5-fold effect of increasing the half-life value of a mesophilic β-glucanase from *Bacillus terquilensis* at 60 °C has been reported for the enzyme mutant forms N31C-T187C and P102C-N125C [22]. In the case of cellulase C (CelC) from *Clostridium thermocellum*, the enzyme activity half-life for the wild-type CelC and its double mutant I232C-N249C at 65 °C were 0.46 and 2.5 min, respectively [28]. Although in our case the achieved effects of increasing the EGLII thermostability were not very high, they are quite important from a practical point of view. The higher specific activity of the enzyme-engineered forms and the better activity retention at 80 °C, that is, a temperature of a pelletizing stage in the production of animal feeds, allows obtaining the final product with improved properties. In particular, a higher β-glucanase activity in barley-based diets helps to mitigate the antinutritive effect of cereal β-glucans on monogastric animals, particularly in poultry, and therefore, using thermostable enzymes in the preparation of feeds attracts the attention of researchers working in this field [40,41].

## 4. Materials and Methods

### 4.1. Disulfide Bond Design

A structure-based design of disulfides was performed through Cys scanning to identify potential mutations that can result in disulfide bonds using Schrödinger’s BioLuminate software [32]. The selection of the most appropriate amino acid pairs for mutations to Cys was decided according to the ΔEi (interaction energy), ΔE (strain energy or the difference between the internal energy before and after relaxation). and the Quality parameter (Good: ΔEi < 0 and ΔStrain E < 20; Bad: ΔEi > 40, ΔStrain E > 40, or ΔEi > 20 and ΔStrain E > 20; Medium: any other values) [32]. Additional criteria for the selection of disulfide bonds includes the distance between C_β_–C_β_ atoms of two amino acids that is not more than 5 Å apart (for glycine: from C_α_), the arrangement of residues on the surface of the protein or between secondary structures, the distance from the catalytic core, the appropriate conformation of amino acids, and the length of disulfide bonds about 2 Å [33,42].

### 4.2. Microbial Strains

The *P. canescens* PCA-10 strain [43] was used as an auxotrophic host strain (*niaD-*) in transformation. The *Escherichia coli* MachI T1^R^ strain (Thermo Fisher Scientific Inc., Waltman, MA, USA) was used to obtain competent cells in the subcloning experiments. 

### 4.3. Enzyme Activity Assays

The enzyme activities toward polysaccharides were determined by analyzing the reducing sugars released after the enzymatic reaction using the Nelson–Somogyi assay [44]. The enzyme activities toward carboxymethylcellulose (CMC), birchwood xylan (Sigma, St. Louis, MO, USA), and barley β-glucan (Megazyme, Boronia, Australia) were assayed at pH 5.0 (0.05 M Na-acetate buffer) and 50 °C using a substrate concentration of 5 mg/mL [45]. 

The enzymatic activities were expressed in international units. One unit of activity corresponded to the quantity of enzyme hydrolyzing 1 µmol of substrate or the release of 1 µmol of reducing sugars (in glucose equivalents) per minute.

### 4.4. Site-Directed Mutagenesis and Protein Expression

Cloning the *P. verruculosum* native *egl2* gene, encoding EGLII, into the *P. canescens* PCA-10 (*niaD*-) auxotrophic strain has been described previously [31]. Six pairs of oligonucleotide primers were constructed to conduct the S12C-A270C, S127C-A165C, and Y171C-L201C substitutions (3 DSBs) in the recombinant EGLII (Table 2).

Mutagenic primers containing the target mutations were designed in the Laboratory of Enzyme Biotechnology (FRC of Biotechnology, RAS) and synthesized in the Syntol laboratory (Moscow, Russia) that specializes in oligonucleotide synthesis. A routine polymerase chain reaction (PCR) was used to perform site-directed mutagenesis. The standard blend of reagents for PCR was supplied by Thermo Fisher Scientific Inc. (Waltman, MA, USA). The isolation of genomic and plasmid DNA as well as the PCR product purification was carried out using QIAGEN Kits (QIAGEN, Valencia, CA, USA). Briefly, in the first round of the PCRs, four DNA fragments for two point mutations were obtained using the genomic DNA of *P. verruculosum* as a template, and then a full-size *egl2* gene (1319 bp), carrying target mutations, was amplified in a second round of PCR with purified fragments as a template. The total number of PCR cycles to introduce two point mutations did not exceed 30. The full-size mutated *egl2* gene was cloned into a modified linear PC1 shuttle vector containing a promoter region of *xylA* gene, encoding xylanase A from *P. canescens*, and a terminator region encoding endoglucanase III from *P. verruculosum* [41]. Next, the resulting expression plasmids were transformed into competent *E. coli* MachI cells, and the produced DNA material was supplied for analysis. The *E. coli* MachI cells were cultured as described elsewhere [46]. Thus, plasmid constructs pPC1-DSB1, pPC1-DSB2, and pPC1-DSB3, containing S12C-A270C, S127C-A165C, and Y171C-L201C substitutions, respectively, were obtained. The absence of additional mutations, deletions, or insertions in the *egl2* gene was confirmed by its sequencing in both directions by the method described by Sanger et al. [47].

The pPC1-DSB1, pPC1-DSB2, and pPC1-DSB3 plasmid constructs were directed into protoplasts of the recipient *P. canescens* PCA-10 (*niaD-*) strain together with a plasmid pSTA10 (10:1, µg) using the modified method described by Aleksenko et al. [48]. The efficiency of integration amounted to 30–40 clones per 1 µg of the target DNA that corresponds to a standard frequency of transformation for the fungal genus *Penicillium* [46]. Stable transformants were grown in flasks for 6 days using a small volume (100 mL) of the liquid culture medium containing 40 g/L soybean hulls, 50 g/L corn extract, and 25 g/L KH_2_PO_4_, and then CMCase and β-glucanase activities were determined in the broth as described above.

### 4.5. Purification and Charactetization of the Thermostable Variants

The purification of EGLII and its mutant forms was carried out as described elsewhere [31]. The isolated enzymes were subjected to a sodium dodecylsulfate polyacrylamide gel electrophoresis (SDS-PAGE) in 12% gel using a Mini Protean II equipment (Bio-Rad Laboratories, Hercules, CA, USA). The staining of protein bands was carried out with Coomassie Blue R-250 (Ferak, Berlin, Germany). The protein concentration in the samples was determined by the modified Lowry method [49], using bovine serum albumin as the standard.

In the thermostability studies, the enzymes were incubated at 70 or 80 °C in a 0.1 M Na-acetate buffer, pH 5.0. At a definite incubation time, an aliquot of the enzyme solution was taken and cooled in cold water, and then the residual activity against β-glucan was assayed under standard conditions as described above. The residual activity was expressed as the percentage of the initial activity. 

### 4.6. Molecular Modelling

In order to understand the role of DSBs in enhancing the enzyme thermostability on a molecular level, classical MD simulations were performed for EGLII-wt and its variants (DSB2 and DSB3) at temperatures 26.85, 70, and 80 °C. Simulations at 26.85 °C were considered to identify the baseline dynamics and a comparison of the structural features of the EGLII-wt and its variants under standard conditions. The temperatures 70 and 80 °C were considered as those used for experimental conditions to observe the structural changes that occured in the thermal unfolding. The starting coordinates of EGLII-wt were taken from the X-ray crystal structure of the enzyme (PDB ID: 5L9C, chain A). The variant structures were generated based on EGLII-wt structures using a FoldX plugin implemented in the YASARA software version 17.8.19 [50,51,52]. For the variants, DSBs were generated using the “interactive SS bridge selection” option implemented in the GROMACS v5.1.2 package [51,52,53,54]. Prior to simulations, water molecules and ligands (cellobiose and N-acetyl-d-glucosamine) present in the crystal structure were removed using the YASARA software version 17.8.19. All MD simulations were performed using the GROMACS v5.1.2 package [53,54,55,56] with the Amber99SB-ILDN force field and extended simple point charge (SPC/E) 216 water model [57,58]. The protonation state of titratable residues was determined based on p*K*_a_ estimation using the PROPKA method implemented in the PDB2PQR server using AMBER99 charges at pH 5 [59,60,61,62]. Specifically, protonation was assigned to Nδ1 for H96, H102, and H207 based on the possible H-bonds networks, and both Nδ1 and Nɛ2 protonated for H144 based on the PROPKA calculation. Subsequently, the enzymes were inserted in a 416.77 nm^3^ cubic box and then explicitly solvated in water, and sodium ions were added to neutralize the charge of the systems. The final systems contained in total approx. 41,050 atoms and 12,190 water molecules. The electrostatic interactions were calculated by applying the particle mesh Ewald (PME) method [63,64]. Short-range electrostatic interactions (rcoulomb) and Van der Waals (rvdw) were calculated using a cutoff value of 1.0 nm each. The energy minimization of each system was performed individually using the steepest descent minimization algorithm until the maximum force reached 1000.0 kJ mol^−1^ nm^−1^. The final system equilibration was conducted in the *NVT* and *NPT* ensembles at three temperatures: 26.85, 70, and 80 °C. The *NVT* and *NPT* equilibrations were performed with a time step of 2 fs at three temperatures for 100 ps. In the case of *NVT* equilibration, initial random velocities were assigned to the atoms of the molecules according to the Maxwell–Boltzmann algorithm at the same temperature. For the *NPT* equilibration, the Berendsen thermostat and Parrinello–Rahman pressure coupling were used to keep the system at a fixed temperature, time constant (τ_T_) of 0.1 ps and 1 bar pressure, and time constant (τ_P_) of 2 ps. All bonds between hydrogen and heavy atoms were constrained with the LINCS algorithm [55,65,66]. The production run was carried out using the *NPT* ensemble for 100 ns with a time step of 2 fs for each temperature. The coordinates were saved every 500 ps from simulation trajectories in the 100 ns simulations. The trajectories and the structures were visualized using VMD 1.9.1 [67]. Analyses including the root mean square deviation (RMSD) of backbone atoms of the protein with respect to a minimized crystal structure [68], the radius of gyration (R_g_) [69], were performed using tools from the GROMACS software package [55]. The stability of -S-S- distance of variants was analyzed using VMD 1.9.1 [67]. 

## 5. Conclusions

Protein engineering based on the DSB design was performed to improve the thermostability of EGLII from *P. verruculosum*. The experimental results showed that the designed variants, DSB2 and DSB3, contributed significantly to the thermostability of EGLII. Simultaneously, the MD simulations revealed that the introduced DSBs rigidified the overall structures and thereby enhanced the thermostability of EGLII. The overall results demonstrate the applicability of the DSB design as an efficient approach towards increasing the thermostability of EGLII that can be applied in similar enzymes. In the growing need of sustainable social development across the world, the application of thermostable enzymes will have significant socioeconomic benefits. In particular, the higher specific activity and the enhanced thermostability of the engineered EGLII should contribute to its more effective use as an additive to animal feeds as well as to its potential application in the processes of the bioconversion of renewable lignocellulosic biomass for the production of a wide variety of fine and bulk chemicals in different biorefineries.

## Figures and Tables

**Figure 1 ijms-20-01602-f001:**
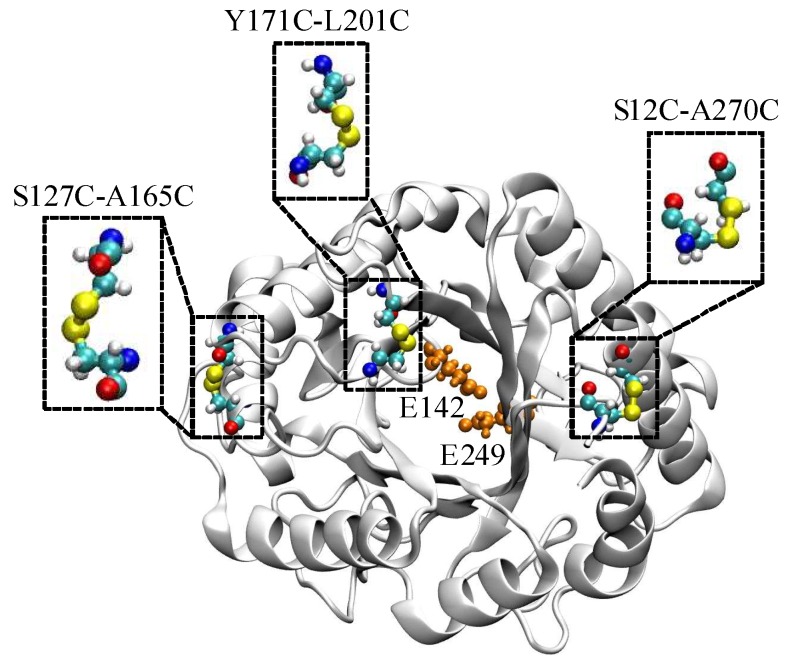
The three potential disulfide bonds in EGLII (PDB ID: 5L9C, chain A) used for the experimental studies: Catalytic residues E142 and E249 in the enzyme active site are shown in orange.

**Figure 2 ijms-20-01602-f002:**
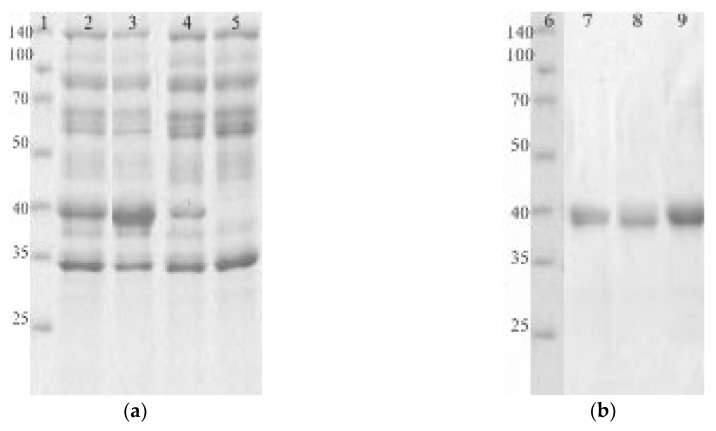
The SDS-PAGEs of culture fluids containing the recombinant EGLII forms (**a**) and purified enzymes (**b**). Lanes 1 and 6 are markers (kDa); lane 2 is the EGLII-DSB2 recombinant strain; lane 3 is the EGLII-DSB3 recombinant strain; lane 4 is the EGLII-wt recombinant strain; lane 5 is the control strain (PCA-10); lane 7 is EGLII-DSB2; lane 8 is EGLII-DSB3; and lane 9 is EGLII-wt.

**Figure 3 ijms-20-01602-f003:**
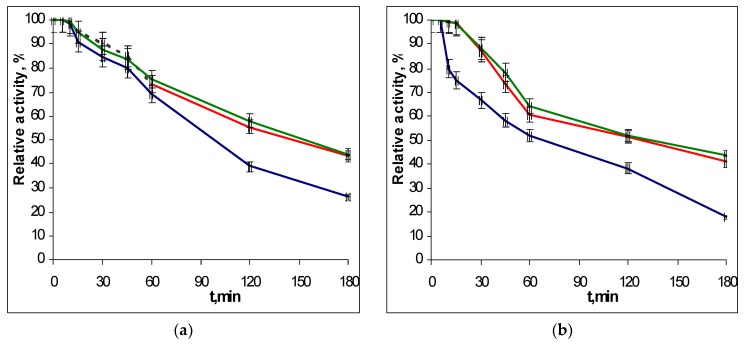
The thermoinactivation of EGLII-wt (blue line) and its DSB2 (red) and DSB3 (green) variants at pH 5.0 and 70 °C: (**a**) The residual activity against CMC and (**b**) the residual activity against β-glucan. The enzyme initial activity is taken as 100%.

**Figure 4 ijms-20-01602-f004:**
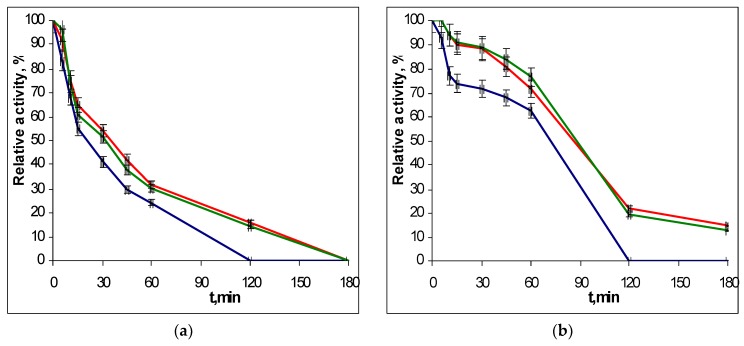
The thermoinactivation of EGLII-wt (blue line) and its DSB2 (red) and DSB3 (green) variants at pH 5.0 and 80 °C: (**a**) Residual activity against CMC and (**b**) the residual activity against β-glucan. The enzyme initial activity is taken as 100%.

**Figure 5 ijms-20-01602-f005:**
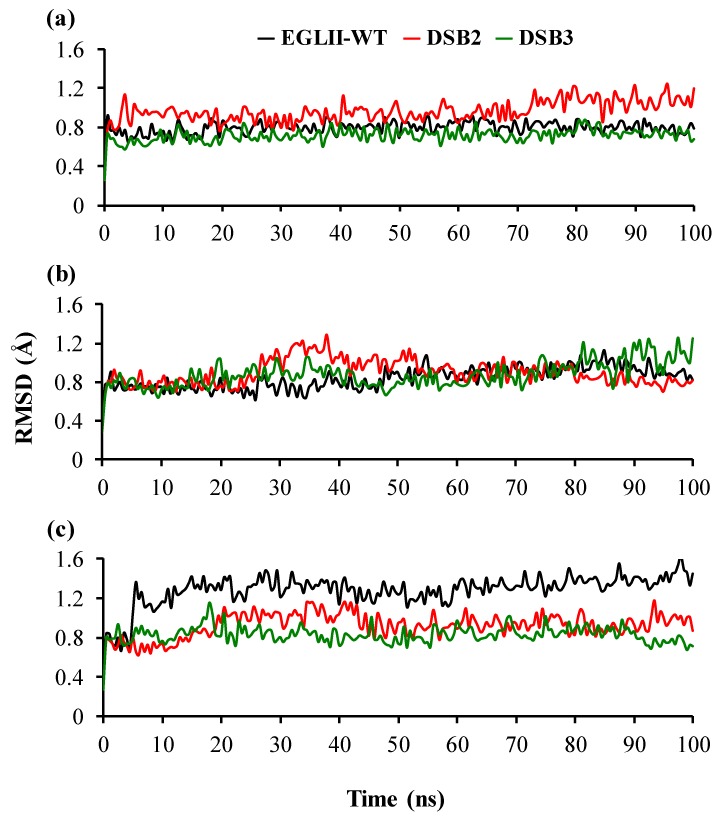
The structural stability of the EGLII forms based on RMSD: (**a**,**b**) At 26.85 and 70 °C, the RMSD of the backbone indicated that the DSB2 and DSB3 variants and EGLII-wt had similar structural stabilities. (**c**) At 80 °C, the RMSD of the backbone of EGLII-wt showed an instability, whereas the structures of the DSB2 and DSB3 variants remained stable.

**Table 1 ijms-20-01602-t001:** The specific activities (U/mg protein) of recombinant EGLII forms.

Substrate	EGLII-wt	EGLII-DSB2	EGLII-DSB3
β-Glucan	57 ± 4	68 ± 6	66 ± 5
CMC	53 ± 3	61 ± 4	64 ± 4

**Table 2 ijms-20-01602-t002:** The sequences of the oligonucleotides.

Mutation	Primer Name	Sequence
S12C	S12C-fwd	5′-AACGTGCTTCTTGTTTCGAATGGTTCGGT-3′
	S12C-rev	5′-ACCGAACCATTCGAAACAAGAAGCACGTT-3′
A270C	A270C-fwd	5′-TGCTGGATTATTTGTGTGAAAACTCAGACGT-3′
	A270C-rev	5′-ACGTCTGAGTTTTCACACAAATAATCCAGC-3′
S127C	S127C-fwd	5′-TGGTCCACACTGGCCTGTCAATTCAAATCA-3′
	S127C-rev	5′-TGATTTGAATTGACAGGCCAGTGTGGACCA-3′
A165C	A165C-fwd	5′-ATGGCATCCGCGACTGTGGTGCAACAA-3′
	A165C-rev	5′-TTGTTGCACCACAGTCGCGGATGCCA-3′
Y171C	Y171C-fwd	5′-TGGCGCAACAACTCAATGTATCTTCGTTGA-3′
	Y171C-rev	5′-TCAACGAAGATACATTGAGTTGTTGCGCCA-3′
L201C	L201C-fwd	5′-ACTGACCCTTCTGATTGTATCGTCTACGAGAT-3′
	L201C-rev	5′-ATCTCGTAGACGATACAATCAGAAGGGTCAGT-3′

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
