# Peer review of "Disulfide Bond Engineering of an Endoglucanase from Penicillium verruculosum to Improve Its Thermostability"

_ijms, 2019, doi:10.3390/ijms20071602_

Round 1

Reviewer 1 Report

Presented manuscript entitled Disulfide bond engineering of an endoglucanase from Penicillium verruculosum to improve its thermostability concerns the structure-based disulfide bond engineering in order to improve the thermostability of endoglucanases from Penicillium verruculosum.

The topic of the presented study could be interesting for a broad variety of readers. Moreover, the manuscript is well-prepared and proposed method of molecular modelling could be used in many applications. In my opinion the presented manuscript could be considered for publication after major revision according to the comments presented below.

In the Introduction section Authors present information about application of endoglucanases and the role of disulfide bonds in enzyme structure. However this part should be improved. For this purpose, see the reference (Catalysts 2017, 7, 374). What is more, in my opinion more data about in silico experiments and its examples should be given in this section.

The novelty of the study should be clearly presented in the manuscript.

In Figure 3 and Figure 4 there are lack of units of relative activity on the Y axis. This information should be provided.

In Figure 5 the description of the Y axis should be imporved for better clarity.

In Table 1 there is the lack of units of specific activities. This information should be provided.

The quality of the Figure S2 presented in Supplementary materials has to be improved.

Units presentation, for example °C should be unified in the whole manuscript.

In the study endoglucanase from Penicillium verruculosum was used. Why this particular subspecies was used to obtain enzyme? Does endoglucanase from Penicillium verruculosu present some advantages as compared to other endoglucanases?

A possible explanation of the differences in the relative activity of EGII-wt after thermoinactivation at 70 °C and 80 °C should be presented. Why the tests were carried out at these two temperatures? Has the investigation of thermoinactivation been carried out of at different temperatures? This information should be provided.

As the topic of the presented study and obtained results are interesting, the Discussion section should be expanded to present more data.

I suggest to expand and improve the Conclusion section.

Please check the whole manuscript to avoid some minors editorial issues.

Please carefully check the References list and improve where necessary to meet the journal standards.

Author Response

We would like to express our sincere gratitude to the anonymous referees for their helpful comments that will help to improve the quality of the manuscript. We have provided below a point-by-point response to referees. We have agreed to all points of the referees and have revised our manuscript accordingly.

Point 1: In the Introduction section Authors present information about application of endoglucanases and the role of disulfide bonds in enzyme structure. However this part should be improved. For this purpose, see the reference (Catalysts 2017, 7, 374). What is more, in my opinion more data about in silico experiments and its examples should be given in this section.

Response 1: We appreciate your valuable suggestion. To provide more examples on in silico experiments for role of disulfide bonds in enzyme structure, following sentences were added in the introduction:

“Recently, DSBs engineering (N31C-T187C/P102C-N125C) of a mesophilic β-glucanase from Bacillus terquilensis showed enhancement of thermostability by 48.3% in increasing half-life at 60°C and a 4.1°C rise in melting temperature compared to wild type [22]».

Point 2: The novelty of the study should be clearly presented in the manuscript.

Response 2: DSBs engineering to improve thermostability remain to be unexplored for endoglucanase II (EGLII) from P. verruculosum. To the best of our knowledge, this study is the first engineering of DSBs for endoglucanases (EGLII from P. verruculosum) to improve the thermostability. In order to highlight the novelty of our work, we have added the following statement in introduction:

“However, DSBs engineering to improve thermostability remain to be unexplored for endoglucanase II (EGLII) from P. verruculosum»

Point 3: In Figure 3 and Figure 4 there are lack of units of relative activity on the Y axis. This information should be provided.

Response 3: We added to the captions for Figures 3 and 4:

«The enzyme initial activity is taken as 100%».

Point 4: In Figure 5 the description of the Y axis should be improved for better clarity.

Response 4: We have revised Figure 5 for better clarity.

Point 5: In Table 1 there is the lack of units of specific activities. This information should be provided.

Response 5: Units of specific activity (U/ mg protein) were added in the heading of Table 1.

Point 6: The quality of the Figure S2 presented in Supplementary materials has to be improved.

Response 6: We have improved the quality of Figure S2.

Point 7: Units presentation, for example C should be unified in the whole manuscript.

Response 7: We checked thoroughly for consistent use of units in the whole manuscript.

Point 8: In the study endoglucanase from Penicillium verruculosum was used. Why this particular subspecies was used to obtain enzyme? Does endoglucanase from Penicillium verruculosum present some advantages as compared to other endoglucanases?

Response 8: We added, at the end of the Introduction, the rationale for the use of EGLII with the addition of a link to an article by Morozova et al., Where it was shown that the enzyme is major and belongs to the number of the most thermostable producer cellulases:
«EGLII is one of major enzymes of P. verruculosum extracellular multienzyme system, and it is an important component of a commercial enzyme preparation Agrocell Plus produced by Agroferment company (Russia) for using as a feed additive. Although EGLII is one of the most thermostable enzymes of P. verruculosum cellulase complex [30], its thermostability is still needed to be improved since the industrial production of the animal feeds involves the pelletizing stage proceeding at ~80 °C»

Point 9: A possible explanation of the differences in the relative activity of EGII-wt after thermoinactivation at 70 Â°C and 80 Â°C should be presented. Why the tests were carried out at these two temperatures? Has the investigation of thermoinactivation been carried out of at different temperatures? This information should be provided.

Response 9: Clarified. Added to section 2.3:

«The choice of conditions for the study of thermal stability was due to the fact that the EGLII is an important component of an industrial enzyme preparation for using as a feed additive (see Introduction), which production involves the pelletizing stage proceeding at ~80 °C».

Point 10: As the topic of the presented study and obtained results are interesting, the Discussion section should be expanded to present more data.

Response 10: Addressed. We have expanded discussion. We compared our results with previous reports.  Following sentences were added in the discussion:

«The half-life times of EGLII at 70 and 80 °C increased by 1.5–2 times as a result of introducing the DSB2 and DSB3. Similar 1.5-fold effect of increasing the half-life value of a mesophilic b-glucanase from Bacillus terquilensis at 60 °C has been reported for the enzyme mutant forms N31C-T187C and P102C-N125C [22]. In the case of cellulase C (CelC) from Clostridium thermocellum, the enzyme activity half-life for the wild-type CelC and its double mutant I232C-N249C at 65 °C was 0.46 and 2.5 min, respectively [28]. Although in our case the achieved effects of increasing the EGLII thermostability were not very high, they are quite important from a practical point of view. The higher specific activity of the enzyme engineered forms and the better activity retention at 80 °C, that is, a temperature of a pelletizing stage in the production of animal feeds, allows obtaining the final product with improved properties. In particular, a higher b-glucanase activity in barley-based diets helps to mitigate the antinutritive effect of cereal b-glucans on monogastric animals, particularly in poultry, and therefore using thermostable enzymes in the preparation of feeds attracts the attention of researchers working in this field [40,41]».

Point 11: I suggest to expand and improve the Conclusion section.

Response 11: Done. We have expanded and improved conclusion section. Following sentences were added in the conclusion:

«Experimental results showed that designed variants, DSB2 and DSB3 contribute significantly to the thermostability of EGLII. Simultaneously, MD simulations revealed that introduced DSBs rigidified overall structures and thereby enhanced thermostability of EGLII. Overall results demonstrate the applicability of the DSB design as an efficient approach towards increasing thermostability of EGLII and can be applied in similar enzymes»

And

«In particular, the higher specific activity and the enhanced thermostability of the engineered EGLII should contribute to its more effective use as an additive to animal feeds as well as to its potential application in the processes of bioconversion of renewable lignocellulosic…»

Point 12: Please check the whole manuscript to avoid some minors editorial issues.

Response 12: Done. We read carefully the manuscript and fixed all editorial issues.

Point 13: Please carefully check the References list and improve where necessary to meet the journal standards.

Response 13: Done. We checked thoroughly the whole manuscript to meet the journal standards.

Reviewer 2 Report

In the proposed manuscript the authors present the results obtained in the increase of thermostability of two variants of EGLII from Penicillium verruculosum, designed using disulphide-bonds engineering, one of the many factors that can improve that complex property. Endoglucanases are important enzymes for cellulosic biomass conversion and other industrial processes and contributions to enhance the thermostability of these enzymes have industrial significance. Obtained results are promising and the study is well conducted.

Author Response

We thank and appreciate the generous comments on our manuscript.

Reviewer 3 Report

This is interesting paper describing application of protein engineering to obtain thermostable endoglucanase. In my opinion the nmuscript could be accepted in present form.

Author Response

(The authors gave the same response as above.)

Round 2

Reviewer 1 Report

Authors have significantly imporved the manuscript after revision. All of the comments have been properly adressed and responded.

I have no further comments. Thus, in my opinion, pmanuscript might be accepted for publication in the present form.